# The Geometry of Deep Networks: Power Diagram Subdivision

**Randall Balestriero, Romain Cosentino, Behnaam Aazhang, Richard G. Baraniuk**
Rice University
Houston, Texas, USA

## Abstract

We study the geometry of deep (neural) networks (DNs) with piecewise affine and convex nonlinearities. The layers of such DNs have been shown to be *max-affine spline operators* (MASOs) that partition their input space and apply a region-dependent affine mapping to their input to produce their output. We demonstrate that each MASO layer's input space partition corresponds to a *power diagram* (an extension of the classical Voronoi tiling) with a number of regions that grows exponentially with respect to the number of units (neurons). We further show that a composition of MASO layers (e.g., the entire DN) produces a progressively subdivided power diagram and provide its analytical form. The *subdivision process* constrains the affine maps on the potentially exponentially many power diagram regions with respect to the number of neurons to greatly reduce their complexity. For classification problems, we obtain a formula for the DN's decision boundary in the input space plus a measure of its curvature that depends on the DN's architecture, nonlinearities, and weights. Numerous numerical experiments support and extend our theoretical results.

## 1   Introduction

Today's machine learning landscape is dominated by deep (neural) networks (DNs), which are compositions of a large number of simple parameterized linear and nonlinear transformations. Deep networks perform surprisingly well in a host of applications; however, surprisingly little is known about why they work so well.

Recently, [BB18a, BB18b] connected a large class of DNs to a special kind of spline, which enables one to view and analyze the inner workings of a DN using tools from approximation theory and functional analysis. In particular, when the DN is constructed using convex and piecewise affine nonlinearities (such as ReLU, Leaky- ReLU, max-pooling, etc.), then its layers can be written as *max-affine spline operators* (MASOs). An important consequence for DNs is that each layer partitions its input space into a set of regions and then processes inputs via a simple affine transformation that changes continuously from region to region. *Understanding the geometry of the layer partition regions – and how the layer partition regions combine into the DN input partition – is thus key to understanding the operation of DNs.*

There has only been limited work in the geometry of deep networks. The originating MASO work of [BB18a, BB18b] focused on the analytical form of the region-dependent affine maps and empirical statistics of the partition without studying the structure of the partition or its construction through depth. The work of [WBB19] empirically studied the partition highlighting the fact that knowledge of the region in which each input lies is sufficient to reach high performance. Other works have focused on the properties of the partition, such as upper bounding the number of regions [MPCB14, RPK+17, HR19]. An explicit characterization of the input space partition of one hidden layer DNs with ReLU activation has been developed in [ZBH+16] by means of tropical geometry.

In this paper, we adopt a computational and combinatorial geometry [PA11, PS12] perspective of MASO-based DNs to derive the analytical form of the input-space partition of a DN unit, a DN layer, and an entire end-to-end DN. Our results apply to any DN employing affine transformations plus piecewise affine and convex nonlinearities.

We summarize our **contributions** as follows: **[C1]** We demonstrate that each MASO DN layer partitions its input (feature map) space according to a *power diagram* (PD) (also known as a Laguerre–Voronoi diagram) [AI] and derive the analytical formula of the PD (Section 3.2). **[C2]** We demonstrate that the composition of the several MASO layers comprising a DN effects a *subdivision* process that creates the overall DN input-space partition and provide the analytical form of the partition (Section 4). **[C3]** We demonstrate how the centroids of the layer PDs can be efficiently computed via backpropagation (Section 4.2), which permits ready visualization of a PD. **[C4]** In the classification setting, we derive an analytical formula for a DN's decision boundary in terms of its input space partition (Section 5). The analytical formula enables us to characterize some key geometrical properties of the boundary.

Our complete, analytical characterization of the input-space and feature map partition of MASO DNs opens up new avenues to study the geometrical mechanisms behind their operation. Additional background information, results, and proofs of the main results are provided in several appendices.

## 2 Background

**Deep Networks.** A deep (neural) network (DN) is an operator $f_\Theta$ with parameters $\Theta$ that maps an input signal $x \in \mathbb{R}^D$ to the output prediction $\widehat{y} \in \mathbb{R}^C$. Current DNs can be written as a composition of $L$ intermediate *layer* mappings $f^{(\ell)} : \mathcal{X}^{(\ell-1)} \to \mathcal{X}^{(\ell)}$ ($\ell = 1, \ldots, L$) with $\mathcal{X}^{(\ell)} \subset \mathbb{R}^{D(\ell)}$ that transform an input *feature map* $z^{(\ell-1)}$ into the output feature map $z^{(\ell)}$ with the initializations $z^{(0)}(x) := x$ and $D(0) = D$. The feature maps $z^{(\ell)}$ can be viewed equivalently as signals, tensors, or flattened vectors; we will use boldface to denote flattened vectors (e.g., $\boldsymbol{z}^{(\ell)}, \boldsymbol{x}$).

DNs can be constructed from a range of different linear and nonlinear operators. One important linear operator is the ***fully connected operator*** that performs an arbitrary affine transformation by multiplying its input by the dense matrix $W^{(\ell)} \in \mathbb{R}^{D(\ell) \times D(\ell-1)}$ and adding the arbitrary bias vector $b_W^{(\ell)} \in \mathbb{R}^{D(\ell)}$ as in $f_W^{(\ell)}\big(z^{(\ell-1)}(\boldsymbol{x})\big) := W^{(\ell)} z^{(\ell-1)}(\boldsymbol{x}) + b_W^{(\ell)}$. Another linear operator is the ***convolution operator*** in which the matrix $W^{(\ell)}$ is replaced with a circulant block circulant matrix denoted as $\boldsymbol{C}^{(\ell)}$. One important nonlinear operator is the ***activation operator*** that applies elementwise a nonlinearity $\sigma$ such as ReLU $\sigma_{\mathrm{ReLU}}(u) = \max(u, 0)$. Further examples are provided in [GBC16]. We define a DN **layer** $f^{(\ell)}$ as a single nonlinear DN operator composed with any (if any) preceding linear operators that lie between it and the preceding nonlinear operator.

**Max Affine Spline Operators (MASOs).** Work from [BB18a, BB18b] connects DN layers with *max-affine spline operators* (MASOs). A MASO is a continuous and convex operator w.r.t. each output dimension $S[A, B] : \mathbb{R}^D \to \mathbb{R}^K$ that concatenates $K$ independent *max-affine splines* [MB09, HD13], with each spline formed from $R$ affine mappings. The MASO parameters consist of the "slopes" $A \in \mathbb{R}^{K \times R \times D}$ and the "offsets/biases" $B \in \mathbb{R}^{K \times R}$.[1] Given the layer input $\boldsymbol{z}^{(\ell-1)}$, a MASO layer produces its output via

$$[\boldsymbol{z}^{(\ell)}(\boldsymbol{x})]_k = \Big[ S[A^{(\ell)}, B^{(\ell)}](\boldsymbol{z}^{(\ell-1)}(\boldsymbol{x})) \Big]_k = \max_{r=1,\ldots,R} \Big( \big\langle [A^{(\ell)}]_{k,r,\cdot}, \boldsymbol{z}^{(\ell-1)}(\boldsymbol{x}) \big\rangle + [B^{(\ell)}]_{k,r} \Big), \quad (1)$$

where $A^{(\ell)}, B^{(\ell)}$ are the per-layer parameters, $[A^{(\ell)}]_{k,r,\cdot}$ represents the vector formed from all of the values of the last dimension of $A^{(\ell)}$, and $[\cdot]_k$ denotes the value of a vector's $k^{\text{th}}$ entry.

The key background result for this paper is that any DN layer $f^{(\ell)}$ constructed from operators that are piecewise-affine and convex can be written as a MASO with parameters $A^{(\ell)}, B^{(\ell)}$ and output dimension $K = D(\ell)$. Hence, a DN is a composition of $L$ MASOs [BB18a, BB18b]. For example, a layer made of a fully connected operator followed by a leaky-ReLU with leakiness $\eta$ has parameters $[A^{(\ell)}]_{k,1,\cdot} = [W^{(\ell)}]_{k,\cdot}, [A^{(\ell)}]_{k,2,\cdot} = \eta[W^{(\ell)}]_{k,\cdot}$ for the slopes and $[B^{(\ell)}]_{k,1,\cdot} = [b^{(\ell)}]_k, [B^{(\ell)}]_{k,2} = \eta[b^{(\ell)}]_k$ for the biases.

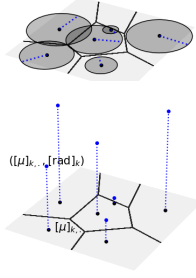

Figure 1: Two equivalent representations of a *power diagram* (PD). **Top:** The grey circles have centroid $[\mu]_{k,\cdot}$ and radii $[\text{rad}]_k$; each point $\boldsymbol{x}$ is assigned to a specific region/cell according to the Laguerre distance from the centroid, which is defined as the length of the segment tangent to and starting on the circle and reaching $\boldsymbol{x}$. **Bottom:** A PD in $\mathbb{R}^D$ (here $D = 2$) is constructed by lifting the centroids $[\mu]_{k,\cdot}$ up into an additional dimension in $\mathbb{R}^{D+1}$ by the distance $[\text{rad}]_k$ and then finding the Voronoi diagram (VD) of the augmented centroids $([\mu]_{k,\cdot}, [\text{rad}]_k)$ in $\mathbb{R}^{D+1}$. The intersection of this higher-dimensional VD with the originating space $\mathbb{R}^D$ yields the PD.

A DN comprising $L$ MASO layers is a non-convex but continuous affine spline operator with an input space partition and a partition-region-dependent affine mapping. However, little is known analytically about the input-space partition. *The goal of this paper is to characterize the geometry of the MASO partitions of the input space and the feature map spaces $\mathfrak{X}^{(\ell)}$.*

**Voronoi and Power Diagrams.** A *power diagram* (PD), also known as a Laguerre–Voronoi diagram [AI], is a generalization of the classical Voronoi diagram (VD).

**Definition 1.** *A PD partitions a space $\mathfrak{X}$ into $R$ disjoint regions/cells $\Omega = \{\omega_1, \ldots, \omega_R\}$ such that $\cup_{r=1}^R \omega_r = \mathfrak{X}$, where each cell is obtained via $\omega_r = \{\boldsymbol{x} \in \mathfrak{X} : r(\boldsymbol{x}) = r\}, r = 1, \ldots, R$, with*

$$r(\boldsymbol{x}) = \underset{k=1,\ldots,R}{\arg\min} \|\boldsymbol{x} - [\mu]_{k,\cdot}\|^2 - [\text{rad}]_k. \tag{2}$$

*The parameter $[\mu]_{k,\cdot}$ is called the* centroid, *while $[\text{rad}]_k$ is called the* radius. *The distance minimized in (2) is called the* Laguerre distance *[IIM85].*

When the radii are equal for all $k$, a PD collapses to a VD. See Fig. 1 for two equivalent geometric interpretations of a PD. For additional insights, see Appendix A and [PS12]. We will have the occasion to use negative radii in our development below. Since $\arg\min_k \|\boldsymbol{x} - [\mu]_{k,\cdot}\|^2 - [\text{rad}]_k = \arg\min_k \|\boldsymbol{x} - [\mu]_{k,\cdot}\|^2 - ([\text{rad}]_k + \rho)$, we can always apply a constant shift $\rho$ to all of the radii to make them positive .

## 3  Input Space Power Diagram of a MASO Layer

Like any spline, it is the interplay between the (affine) spline mappings and the input space partition that work the magic in a MASO DN. Indeed, the partition opens up new geometric avenues to study how a MASO-based DN clusters and organizes signals in a hierarchical fashion.

*We now embark on a programme to fully characterize the geometry of the input space partition of a MASO-based DN. We will proceed in three steps by studying the partition induced by i) one unit of a single DN layer (Section 3.1), ii) the combination of all units in a single layer (Section 3.2), iii) the composition of $L$ layers that forms the complete DN (Section 4).*

### 3.1  MAS Unit Power Diagram

A MASO layer combines $K$ max affine spline (MAS) units $z_k(\boldsymbol{x})$ to produce the layer output $\boldsymbol{z}(\boldsymbol{x}) = [z_1(\boldsymbol{x}), \ldots, z_K(\boldsymbol{x})]^T$ given an input $\boldsymbol{x} \in \mathfrak{X}$. To streamline our argument, we omit the $\ell$ superscript and denote the layer input by $\boldsymbol{x}$. Denote each MAS computation from (1) as

$$z_k(\boldsymbol{x}) = \max_{r=1,\ldots,R} \langle [A]_{k,r,\cdot}, \boldsymbol{x} \rangle + [B]_{k,r} = \max_{r=1,\ldots,R} \mathcal{E}_{k,r}(\boldsymbol{x}), \tag{3}$$

where $\mathcal{E}_{k,r}(\boldsymbol{x})$ is the affine projection of $\boldsymbol{x}$ parameterized by the slope $[A]_{k,r,\cdot}$ and offset $[B]_{k,r}$. By defining the following half-space consisting of the set of points above the hyperplane

$$\mathcal{E}_{k,r}^+ = \{(\boldsymbol{x}, y) \in \mathfrak{X} \times \mathbb{R} : y \geq \mathcal{E}_{k,r}(\boldsymbol{x})\}, \tag{4}$$

we obtain the following geometric interpretation of the unit output.

**Proposition 1.** *The $k^{th}$ MAS unit maps its input space onto the boundary of the convex polytope $\mathcal{P}_k = \cap_{r=1}^R \mathcal{E}_{k,r}^+$, leading to*

$$\{(\boldsymbol{x}, z_k(\boldsymbol{x})), \boldsymbol{x} \in \mathfrak{X}\} = \partial \mathcal{P}_k, \tag{5}$$

*where $\partial \mathcal{P}_k$ denotes the boundary of the polytope.*

The MAS computation can be decomposed geometrically as follows. The slope $[A]_{k,r,\cdot}$ and offset $[B]_{k,r}$ parameters describe the shape of the half-space $\mathcal{E}_{k,r}^+$. The $\max$ over the regions $r$ in (3) defines the polytope $\mathcal{P}_k$ as the intersection over the $R$ half-spaces. The following property shows how the unit projection, the polytope faces and the unit input space partition naturally tie together.

**Lemma 1.** *The vertical projection on the input space $\mathcal{X}$ of the faces of the polytope $\mathcal{P}_k$ from (5) define the cells of a PD.*

Furthermore, we can highlight the maximization process of the unit computation (3) with the following operator $r_k : \mathcal{X} \to \{1, \ldots, R\}$ defined as

$$r_k(\boldsymbol{x}) = \underset{r=1,\ldots,R}{\arg\max}\, \mathcal{E}_{k,r}(\boldsymbol{x}). \tag{6}$$

This operator keeps track of the index of the affine mapping used to produce the unit output or, equivalently, the index of the polytope face used to produce the unit output. The collection of inputs having the same face allocation, defined as $\forall r \in \{1, \ldots, R\}\,, \omega_r = \{\boldsymbol{x} \in \mathcal{X} : r_k(\boldsymbol{x}) = r\}$, constitutes the $r^{\text{th}}$ *partition cell* of the unit $k$ PD (recall (2) and Lemma 1).

The polytope formulation of a DN's PD provides an avenue to study the interplay between the slope and offset of the MAS unit and this specific partition by providing the analytical form of the PD.

**Theorem 1.** *The $k^{th}$ MAS unit partitions its input space according to a PD with $R$ centroids and radii given by $[\mu]_{k,r} = [A]_{k,r,\cdot}$ and $[\text{rad}]_{k,r} = 2[B]_{k,r} + \|[A]_{k,r,\cdot}\|^2, \forall r \in \{1, \ldots, R\}$ (recall (2)).*

**Corollary 1.** *The input space partition of a DN unit is composed of convex polytopes.*

For a single MAS unit, the slope corresponds to the centroid, and its $\ell_2$ norm combines with the bias to produce the radius. The PD simplifies to a VD when $[B]_{k,r} = -\frac{1}{2}\|[A]_{k,r,\cdot}\|^2 + c, \forall r, \forall c \in \mathbb{R}$.

### 3.2 MASO Layer Power Diagram

We study the input space partition of an entire DN layer by studying the joint behavior of all its constituent units. A MASO layer is a continuous, piecewise affine operator made by the concatenation of $K$ MAS units (recall (1)); we extend (3) to

$$\boldsymbol{z}(\boldsymbol{x}) = \left[\max_{r=1,\ldots,R} \mathcal{E}_{1,r}(\boldsymbol{x}), \ldots, \max_{r=1,\ldots,R} \mathcal{E}_{K,r}(\boldsymbol{x})\right]^T, \quad \forall \boldsymbol{x} \in \mathcal{X} \tag{7}$$

and the per-unit face index function $r_k$ (6) into the operator $\boldsymbol{r} : \mathcal{X} \to \{1, \ldots, R\}^K$ defined as

$$\boldsymbol{r}(\boldsymbol{x}) = [\boldsymbol{r}_1(\boldsymbol{x}), \ldots, \boldsymbol{r}_K(\boldsymbol{x})]^T. \tag{8}$$

Following the geometric interpretation of the unit output from Proposition 1, we extend (4) to

$$E_{\boldsymbol{r}}^+ = \left\{(\boldsymbol{x}, \boldsymbol{y}) \in \mathcal{X} \times \mathbb{R}^K : [\boldsymbol{y}]_1 \geq \mathcal{E}_{1,[\boldsymbol{r}]_1}(\boldsymbol{x}), \ldots, [\boldsymbol{y}]_K \geq \mathcal{E}_{K,[\boldsymbol{r}]_K}(\boldsymbol{x})\right\}, \forall \boldsymbol{r} \in \{1, \ldots, R\}^K \tag{9}$$

in order to provide the following layer output geometrical interpretation.

**Proposition 2.** *The layer operator $\boldsymbol{z}$ maps its input space into the boundary of the $\dim(\mathcal{X}) + K$ dimensional convex polytope $\mathbf{P} = \bigcap_{\boldsymbol{r} \in \{1,\ldots,R\}^K} E_{\boldsymbol{r}}^+$ via*

$$\partial\mathbf{P} = \{(\boldsymbol{x}, \boldsymbol{z}(\boldsymbol{x})), \forall \boldsymbol{x} \in \mathcal{X}\}. \tag{10}$$

Similarly to Proposition 1, the polytope $\mathbf{P}$ imprints the layer's input space with a partition that is the intersection of the $K$ per-unit input space partitions.

**Lemma 2.** *The vertical projection on the input space $\mathcal{X}$ of the faces of the polytope $\mathbf{P}$ from Proposition 2 define the cells of a PD.*

The MASO layer projects an input $\boldsymbol{x}$ onto the polytope face indexed by $\boldsymbol{r}(\boldsymbol{x})$ corresponding to

$$\boldsymbol{r}(\boldsymbol{x}) = \left[\underset{r=1,\ldots,R}{\arg\max}\, \mathcal{E}_{1,r}(\boldsymbol{x}), \ldots, \underset{r=1,\ldots,R}{\arg\max}\, \mathcal{E}_{K,r}(\boldsymbol{x})\right]^T. \tag{11}$$

The collection of inputs having the same face allocation jointly across the $K$ units constitutes the $\boldsymbol{r}^{\text{th}}$ *partition cell* (region) of the layer PD.

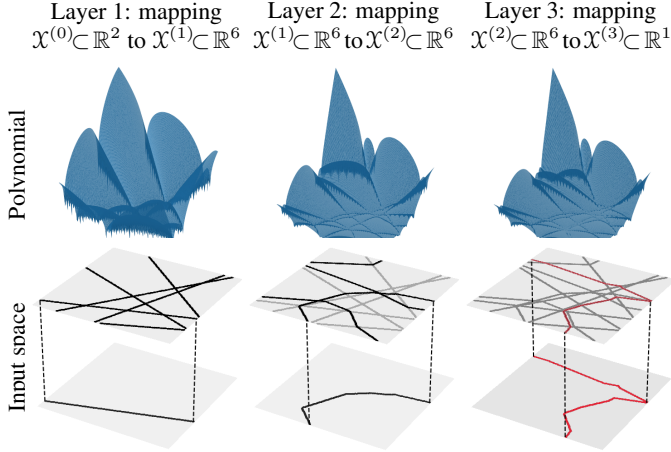

Layer 1: mapping $\mathcal{X}^{(0)} \subset \mathbb{R}^2$ to $\mathcal{X}^{(1)} \subset \mathbb{R}^6$

Layer 2: mapping $\mathcal{X}^{(1)} \subset \mathbb{R}^6$ to $\mathcal{X}^{(2)} \subset \mathbb{R}^6$

Layer 3: mapping $\mathcal{X}^{(2)} \subset \mathbb{R}^6$ to $\mathcal{X}^{(3)} \subset \mathbb{R}^1$

Figure 2: Power diagram subdivision in a toy deep network (DN) with a $D = 2$ dimensional input space. **Top:** The partition polynomial (22), whose roots define the partition boundaries in the input space. **Bottom:** Evolution of the input space partition (15) displayed layer by layer, with the newly introduced boundaries in darker color. Below each partition, one of the newly introduced cuts $\text{edge}_{\mathcal{X}^{(0)}}(k, \ell)$ from (21) is highlighted; in the final layer (right), this cut corresponds to the decision boundary (in red).

**Theorem 2.** *A DN layer partitions its input space according to a PD containing up to $R^K$ cells with centroids $\mu_{\boldsymbol{r}} = \sum_{k=1}^{K} [A]_{k,[\boldsymbol{r}]_k}$, and radii $\text{rad}_{\boldsymbol{r}} = 2\sum_{k=1}^{K} [B]_{k,[\boldsymbol{r}]_k} + \|\mu_{\boldsymbol{r}}\|^2$ (recall (2)).*

**Corollary 2.** *The input space partition of a DN layer is composed of convex polytopes.*

Extending Theorem 1, we observe in the layer case that the centroid of each PD cell corresponds to the sum of the rows of the slopes matrix producing the layer output. The radii involve the bias units and the $\ell_2$ norm of the slopes as well as their correlation. This highlights how, even when a change of weight occurs for a single unit, it will impact multiple centroids and hence multiple cells. Note also that orthogonal DN filters [2] and $[B]_{k,r} = -\frac{1}{2}\|[A]_{k,r,\cdot}\|^2$ reduces the PD to a VD.

Appendix A.2 explores how the shapes and orientations of layer's PD cells can be designed by appropriately constraining the values of the DN's weights and biases.

## 4 Input Space Power Diagram of a MASO Deep Network

We are now armed to characterize and study the input space partition of an entire DN by studying the joint behavior of its constituent layers.

### 4.1 The Power Diagram Subdivision Recursion

We provide the formula for the input space partition of an $L$-layer DN by means of a recursion. Recall that each layer partitions its input space $\mathcal{X}^{(\ell-1)}$ in terms of the polytopes $\mathbf{P}^{(\ell)}$ according to Proposition 2. The DN partition corresponds to a recursive subdivision where each per-layer polytope subdivides the previously obtained partition.

**Initialization ($\ell = 0$):** Define the region of interest in the input space $\mathcal{X}^{(0)} \subset \mathbb{R}^D$.

**First step ($\ell = 1$):** The first layer subdivides $\mathcal{X}^{(0)}$ into a PD via Theorem 2 with parameters $A^{(1)}, B^{(1)}$ to obtain the layer-1 partition $\Omega^{(1)}$.

**Recursion step ($\ell = 2$):** For concreteness we focus here on how the second layer subdivides the first layer's input space partition. In particular, we highlight how a single cell $\omega_{\boldsymbol{r}^{(1)}}^{(1)}$ of $\Omega^{(1)}$ is subdivided, the same applies to all the cells. On this cell, the first layer mapping is affine with parameters $A_{\boldsymbol{r}^{(1)}}^{(1)}, B_{\boldsymbol{r}^{(1)}}^{(1)}$. This convex cell thus remains a convex cell at the output of the first layer mapping, it lives in $\mathcal{X}^{(1)}$ and it is defined as

$$\text{aff}_{\boldsymbol{r}^{(1)}} = \left\{ A_{\boldsymbol{r}^{(1)}}^{(1)} \boldsymbol{x} + B_{\boldsymbol{r}^{(1)}}^{(1)}, \boldsymbol{x} \in \omega_{\boldsymbol{r}^{(1)}}^{(1)} \right\} \subset \mathcal{X}^{(1)}. \tag{12}$$

The second layer partitions its input space $\mathcal{X}^{(1)}$ and thus also potentially subdivisions $\text{aff}_{\boldsymbol{r}^{(1)}}$. In particular, this -mapped cell- will be subdivided by the edges of the polytope $\mathbf{P}^{(2)}$ (recall (10)) having for domain $\text{aff}_{\boldsymbol{r}^{(1)}}$, this domain restricted polytope is defined as

$$\mathbf{P}_{\boldsymbol{r}^{(1)}}^{(2)} = \mathbf{P}^{(2)} \cap \left( \text{aff}_{\boldsymbol{r}^{(1)}} \times \mathbb{R}^{D(2)} \right). \tag{13}$$

Since the layer 1 mapping is affine in this region, the domain restricted polytope $\mathbf{P}^{(2)}_{\boldsymbol{r}^{(1)}}$ can be expressed as part of $\mathcal{X}^{(0)}$ as opposed to $\mathcal{X}^{(1)}$.

**Definition 2.** *The domain restricted polytope $\mathbf{P}^{(2)}_{\boldsymbol{r}^{(1)}} \in \mathcal{X}^{(1)} \times \mathbb{R}^{D(2)}$ can be expressed in $\mathcal{X}^{(0)} \times \mathbb{R}^{D(2)}$ as*

$$\mathbf{P}^{(1\leftarrow2)}_{\boldsymbol{r}^{(1)}} = \cap_{\boldsymbol{r}^{(2)}} \left\{ (\boldsymbol{x}, \boldsymbol{y}) \in \omega^{(1)}_{\boldsymbol{r}^{(1)}} \times \mathbb{R}^{D(1)} : [\boldsymbol{y}]_1 \geq \mathcal{E}^{(1\leftarrow2)}_{1,[\boldsymbol{r}^{(2)}]_1}(\boldsymbol{x}), \ldots, [\boldsymbol{y}]_{D(1)} \geq \mathcal{E}^{(1\leftarrow2)}_{D(1),[\boldsymbol{r}^{(2)}]_{D(1)}}(\boldsymbol{x}) \right\} \quad (14)$$

*with $\mathcal{E}^{(1\leftarrow2)}_{k,[\boldsymbol{r}^{(1)}]_k}$ the hyperplane with slope $A^{(1)}_{\boldsymbol{r}^{(1)}}{}^T A^{(2)}_{\boldsymbol{r}^{(2)}}$ and bias $\left\langle [A^{(2)}_{\boldsymbol{r}^{(2)}}]_{k,r,.}, B^{(1)}_{\boldsymbol{r}^{(1)}} \right\rangle + B^{(2)}_{\boldsymbol{r}^{(2)}}, k \in \{1, \ldots, D(1)\}$.*

The above results demonstrates how cell $\omega^{(1)}_{\boldsymbol{r}^{(1)}}$, seen as $\text{aff}_{\boldsymbol{r}^{(1)}}$ by the second layer, is subdivided by the domain restricted polytope $\mathbf{P}^{(2)}_{\boldsymbol{r}^{(1)}}$; and conversely, how this subdivision of $\omega^{(1)}_{\boldsymbol{r}^{(1)}}$ is done by the domain restricted second layer polytope expressed in the DN input space $\mathbf{P}^{(1\leftarrow2)}_{\boldsymbol{r}^{(1)}}$. Now, combining the latter interpretation, and applying Lemma 2, we obtain that this cell is subdivided according to a PD induced by the faces of $\mathbf{P}^{(1\leftarrow2)}_{\boldsymbol{r}^{(1)}}$, denoted as $\text{PD}^{(1\leftarrow2)}_{\boldsymbol{r}^{(1)}}$. This PD is characterized by the centroids $\mu^{(1\leftarrow2)}_{\boldsymbol{r}^{(1)},\boldsymbol{r}^{(2)}} = A^{(1)}_{\boldsymbol{r}^{(1)}}{}^\top \mu^{(1\leftarrow2)}_{\boldsymbol{r}^{(2)}}$, and radii $\text{rad}^{(1\leftarrow2)}_{\boldsymbol{r}^{(1)},\boldsymbol{r}^{(2)}} = \|\mu^{(1\leftarrow2)}_{\boldsymbol{r}^{(1)},\boldsymbol{r}^{(2)}}\|^2 + 2\langle \mu^{(2)}_{\boldsymbol{r}^{(2)}}, B^{(1)}_{\boldsymbol{r}^{(1)}}\rangle + 2\langle \mathbf{1}, B^{(2)}_{\boldsymbol{r}^{(2)}}\rangle, \forall \boldsymbol{r}^{(2)} \in \{1, \ldots, R\}^{D(2)}$. The PD parameters thus combine the affine parameters $A^{(1)}_{\boldsymbol{r}^{(1)}}, B^{(1)}_{\boldsymbol{r}^{(1)}}$ of the considered cell with the second layer parameters $A^{(2)}, B^{(2)}$. Repeating this subdivision process for all cells $\omega^{(1)}_{\boldsymbol{r}^{(1)}}$ from $\Omega^{(1)}$ forms the subdivided input space partition $\Omega^{(1,2)} = \cup_{\boldsymbol{r}^{(1)}} \text{PD}^{(1\leftarrow2)}_{\boldsymbol{r}^{(1)}}$.

**Recursion step** ($\ell$): Consider the situation at layer $\ell$ knowing $\Omega^{(1,\ldots,\ell-1)}$ from the previous subdivision steps. Similarly to the $\ell = 2$ step, layer $\ell$ subdivides each cell in $\Omega^{(1,\ldots,\ell-1)}$ to produce $\Omega^{(1,\ldots,\ell)}$ leading to the up-to-layer-$\ell$-layer DN partition

$$\Omega^{(1,\ldots,\ell)} = \cup_{\boldsymbol{r}^{(1)},\ldots,\boldsymbol{r}^{(\ell-1)}} \text{PD}^{(1\leftarrow\ell)}_{\boldsymbol{r}^{(1)},\ldots,\boldsymbol{r}^{(\ell-1)}}. \quad (15)$$

See Fig. 2 for a numerical example with a 3-layer DN and $D = 2$ dimensional input space. (See also Figures 7 and 9 in Appendix B.)

**Theorem 3.** *Each cell $\omega^{(1,\ldots,\ell-1)}_{\boldsymbol{r}^{(1)},\ldots,\boldsymbol{r}^{(\ell-1)}} \in \Omega^{(1,\ldots,\ell-1)}$ is subdivided into $\text{PD}^{(1\leftarrow\ell)}_{\boldsymbol{r}^{(1)},\ldots,\boldsymbol{r}^{(\ell-1)}}$, a PD with domain $\omega^{(1,\ldots,\ell-1)}_{\boldsymbol{r}^{(1)},\ldots,\boldsymbol{r}^{(\ell-1)}}$ and parameters*

$$\mu^{(1\leftarrow\ell)}_{\boldsymbol{r}^{(1)},\ldots,\boldsymbol{r}^{(\ell)}} = (A^{(1\leftarrow\ell-1)}_{\boldsymbol{r}^{(1)},\ldots,\boldsymbol{r}^{(\ell-1)}})^T \mu^{(\ell)}_{\boldsymbol{r}^{(\ell)}} \qquad \text{(centroids)} \quad (16)$$

$$\text{rad}^{(1\leftarrow\ell)}_{\boldsymbol{r}^{(1)},\ldots,\boldsymbol{r}^{(\ell)}} = \|\mu^{(1\leftarrow\ell)}_{\boldsymbol{r}^{(1)},\ldots,\boldsymbol{r}^{(\ell)}}\|^2 + 2\langle \mu^{(\ell)}_{\boldsymbol{r}^{(\ell)}}, B^{(1\rightarrow\ell-1)}_{\boldsymbol{r}^{(1)},\ldots,\boldsymbol{r}^{(\ell-1)}}\rangle + 2\langle \boldsymbol{I}, B^{(\ell)}_{\boldsymbol{r}^{(\ell)}}\rangle \qquad \text{(radii)}, \quad (17)$$

*$\forall \boldsymbol{r}^{(i)} \in \{1, \ldots, R\}^{D(i)}$ with $B^{(1\rightarrow\ell-1)} = \sum_{\ell'=1}^{\ell-1} \left( \prod_{i=\ell-1}^{\ell'} A^{(i)}_{\boldsymbol{r}^{(i)}} \right) B^{(\ell')}_{\boldsymbol{r}^{(\ell')}}$ forming $\Omega^{(1,\ldots,\ell)}$.*

The subdivision recursion provides a direct result on the shape of the DN input space partition regions.

**Corollary 3.** *For any number of MASO layers $L \geq 1$, the PD cells of the DN input space partition are convex polytopes.*

## 4.2 Centroid and Radius Computation

While in general a DN has a tremendous number of PD cells, the DN's forward inference calculation locates the cell containing an input signal $\boldsymbol{x}$ with a computational complexity that is only logarithmic in the number of regions. (See Appendix A.3 for a proof and additional discussion.) We now produce a closed-form formula for the radius and centroid of that cell.

Consider the cell of the PD induced by layers 1 through $\ell$ of a DN that contains a data point $\boldsymbol{x}$ of interest. This cell is described by the code $\boldsymbol{r}^{(1)}(\boldsymbol{x}), \ldots, \boldsymbol{r}^{(\ell)}(\boldsymbol{x})$ that we will simplify here in an abuse of notation so simply $\boldsymbol{x}$. Denote the Jacobian operator as $\mathbf{J}$, and the vector of ones by $\mathbf{1}$, the centroid and radius of the cell are given by

$$\mu^{(1\leftarrow\ell)}_{\boldsymbol{x}} = (\mathbf{J}_{\boldsymbol{x}} f^{(1\rightarrow\ell)})^T \mathbf{1}, \quad (18)$$

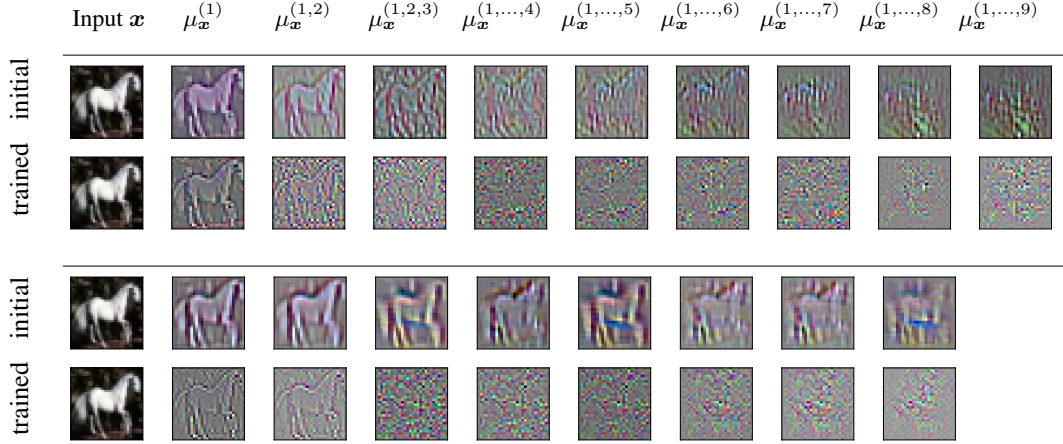

Figure 3: Centroids of the PD regions containing an input horse image $x$ computed via (18) for a LargeConv network (top) and a ResNet (bottom). (See Fig. 11 for results with a SmallConv network.) The input belongs to the PD cell $\omega_x^{(1,\dots,\ell)}$ for each successively refined PD subdivision of each layer $\Omega^{(1,\dots,\ell)}$. At each layer of the subdivision, the region has an associated centroid $\mu_x^{(1,\dots,\ell)}$ (depicted here) and radius (not depicted). As the depth $\ell$ increases, the centroids diverge from horse-like images. This is because the radii begin to dominate the centroids, pushing the centroids outside the PD cell containing $x$. Training accelerates this domineering effect.

$$\operatorname{rad}_x^{(1\leftarrow\ell)} = \|\mu_x^{(1\leftarrow\ell)}\|^2 + 2\left\langle \mathbf{1}, B_x^{(\ell)}\right\rangle + 2\left\langle f^{(1\to\ell)}(x) - A_x^{(1\to\ell-1)}x, \sum_{k=1}^{D(\ell)}[A_x^{(\ell)}]_{k,\cdot}\right\rangle \tag{19}$$

with $A_x^{(1\to\ell-1)} = \left(\nabla_x f_1^{(1\to\ell-1)}, \dots, \nabla_x f_{D(\ell)}^{(1\to\ell-1)}\right)^T$, and where we recall that $\mu_x^{(\ell)} = \sum_{k=1}^{D(\ell)}[A_x^{(\ell)}]_{k,\cdot}$, $B_x^{(1\to\ell-1)} = f^{(1\to\ell)}(x) - A_x^{(1\to\ell-1)}x$ from Theorem 3 and $f_k^{(1\to\ell)}$ is the $k^{\text{th}}$ unit of the layer 1 to $\ell$ mapping. Note how the centroids and biases of the current layer are mapped back to the input space $\mathfrak{X}^{(0)}$ via a projection onto the tangent hyperplane defined by the basis $A_x^{(1\to\ell-1)}$.

Conveniently, the centroids (18) can be computed via an efficient backpropagation pass through the DN, which is typically available because it is integral to DN learning. Moreover, (18) corresponds to the element-wise summation of the *saliency maps* [SVZ13, ZF14] from all of the layer units.[3] Figure 3 visualizes the centroids of the cell containing a particular input signal for a LargeConv and ResNet DN trained on the CIFAR10 dataset (see Appendix C for details on the models plus additional figures).

### 4.3 Distance to the Nearest PD Cell Boundary

In Appendix D we derive the Euclidean distance from a data point $x$ to the nearest boundary of its PD cell (a point from $\partial\Omega$)

$$\min_{u\in\partial\Omega}\|x-u\| = \min_{\ell=1,\dots,L}\min_{k=1,\dots,D(\ell)}\frac{|(z_k^{(\ell)}\circ\cdots\circ z^{(1)})(x)|}{\|\nabla_x(z_k^{(\ell)}\circ\cdots\circ z^{(1)})(x)\|}. \tag{20}$$

Fig. 4 (and 6 in the Appendix) plots the distributions of the log distances from the training points in the CIFAR10 training set to their nearest region boundary the input space partition as a function of layer $\ell$ and at different stages of learning. We see that training increases the number of data points that lie close to their nearest boundary. We see from these figures that while a network with fully connected layers (MLP) refines its partition by introducing cuts close to the training points at each layer, the SmallCNN does not reduce the shortest distance at deeper layers.

A further exploration is carried out in Appendix A.4, where Table 1 summarizes the performance of the centroids, when used as centroids of a VD, to recover inside their region, the same input as the one that originally produced the centroid.

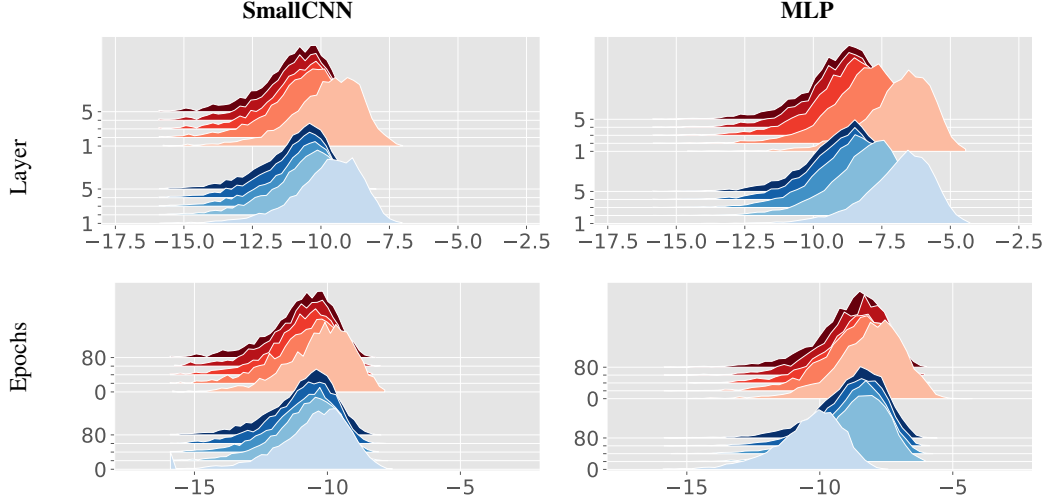

Figure 4: Empirical distributions of the log distances from the training points of the CIFAR10 dataset to the nearest PD cell boundary as calculated by (20) for the various layers of a SmallCNN (left) and MLP (right). Blue: Training set. Red: Test set. On **top** is the evolution through layers at the end of the training, on **bottom** is the evolution of the last layer, through the epochs. The distances decrease with $\ell$ due to PD subdivision which reduces the volume of the cells as the subdivision process occurs. The distances are also much smaller for the CNN desptie having the same number of units for the MLP as the number of filters and translations for the convolutional layers. This demonstrates how the subdivision process of the convolutional layer is much more performance at refining the DN input space partitioning around the data for image data.

## 5 Geometry of the Deep Network Decision Boundary

We now study the edges of the polytopes that define the PD cells' boundaries. We demonstrate how a single unit at layer $\ell$ defines multiple cell boundaries in the input space and use this finding to derive an analytical formula for the DN decision boundary that would be used in a classification task. Without loss of generality, we focus in this section on piecewise nonlinearities with $R = 2$, such as ReLU, leaky-ReLU, and absolute value.

### 5.1 Partition Boundaries and Edges

In the case of $R = 2$ nonlinearities, the polytope $\mathcal{P}_k^{(\ell)}$ of unit $z_k^{(\ell)}$ contains a single edge, we consider here nonlinearities that can be expressed as a leaky-ReLU with leakiness $\eta \neq 0$. We define this edge as the intersection of the faces of the polytope. For instance, in the case of leaky-ReLU, the polytope contains two faces that characterize the two regions produced by a single leaky-ReLU unit. We formally define the edge of a polytope as follows.

**Definition 3.** *The edges of the polytope $\mathcal{P}_k^{(\ell)}$ can be expressed in any space $\mathfrak{X}^{(\ell')}, \ell' < \ell$ (and in particular the input space $\mathfrak{X}^{(0)}$) as*

$$\text{edge}_{\mathfrak{X}^{(\ell')}}(k, \ell) = \{\boldsymbol{x} \in \mathfrak{X}^{(\ell')} : \mathcal{E}_{k,2}^{(\ell)}(\boldsymbol{z}^{(\ell' \to \ell-1)}(\boldsymbol{x})) = 0\}, \tag{21}$$

*with $\boldsymbol{z}^{(\ell' \to \ell-1)} = \boldsymbol{z}^{(\ell-1)} \circ \cdots \circ \boldsymbol{z}^{(\ell')}$, $\mathcal{E}_{k,2}^{(\ell)}$ from (3), and where $\circ$ denotes the composition operator.*

In the same way that the polytopes $\mathbf{P}_{\boldsymbol{r}^1, \ldots, \boldsymbol{r}^{(\ell-1)}}^{(1 \leftarrow \ell)}$ could be expressed in $\mathfrak{X}^{(0)} \times \mathbb{R}^{D(\ell)}$ and then mapped to the DN input space (recall Section 4.1), these edges defined in $\mathfrak{X}^{(\ell-1)}$ can be expressed in the DN input space $\mathfrak{X}^{(0)}$. The projection of the edges into the DN input space will constitute the partition boundaries. Defining the polynomial

$$\text{Pol}(\boldsymbol{x}) = \prod_{\ell=1}^{L} \prod_{k=1}^{D(\ell)} (z_k^{(\ell)} \circ \boldsymbol{z}^{(\ell-1)} \circ \cdots \circ \boldsymbol{z}^{(1)})(\boldsymbol{x}), \tag{22}$$

we obtain the following result where the boundaries of $\Omega^{(1, \ldots, \ell)}$ from Theorem 3 can be expressed in term of the polytope edges and roots of the polynomial.

**Theorem 4.** *The polynomial (22) is of order $\prod_{\ell=1}^{L} D(\ell)$, and its roots correspond to the partition boundaries:*

$$\partial\Omega^{(1,\dots,\ell)} = \{\boldsymbol{x} \in \mathcal{X}^{(0)} : \mathrm{Pol}(\boldsymbol{x}) = 0\} = \cup_{\ell'=1}^{\ell} \cup_{k=1}^{D(\ell')} \mathrm{edge}_{\mathcal{X}^{(0)}}(k, \ell). \tag{23}$$

*The root order defines the dimensionality of the root (boundary, corner, etc.).*

### 5.2 Decision Boundary Curvature

The final DN layer introduces a last subdivision of the partition. For brevity, we focus on a binary classification problem; in this case, $D^{(L)} = 1$ and a single last subdivision occurs, leading to the class prediction being $y = \mathbb{1}_{z_1^{(L)}(\boldsymbol{x}) > \tau}$ for some threshold $\tau$, this last layer can thus be cast as a MASO with a leaky-ReLU type nonlinearity with proper bias, and setting $\tau = 0$. That is, the DN prediction is unchanged by this last nonlinearity, and the change of sign is the change of class is the decision boundary.

**Proposition 3.** *The decision boundary of a DN with $L$ layers is the edge of the last layer polytope $\mathbf{P}^{(L)}$ expressed in the input space $\mathcal{X}^{(0)}$ from Definition 3 as*

$$\mathrm{DecisionBoundary} = \{\boldsymbol{x} \in \mathcal{X}^{(0)} : f(\boldsymbol{x}) = 0\} = \mathrm{edge}_{\mathcal{X}^{(0)}}(1, L), \tag{24}$$

*where $\mathrm{edge}_{\mathcal{X}^{(0)}}(1, L)$ denotes the edge of unit 1 of layer $L$ expressed in the input space $\mathcal{X}^{(0)}$.*

To provide insights into this result, consider a 3-layer DN denoted as $f$ and a binary classification task; we have

$$\mathrm{DecisionBoundary} = \cup_{\boldsymbol{r}^{(2)}} \cup_{\boldsymbol{r}^{(1)}} \{x \in \mathcal{X}^{(0)} : \langle \alpha_{\boldsymbol{r}^{(2)}, \boldsymbol{r}^{(1)}}, \boldsymbol{x} \rangle + \beta_{\boldsymbol{r}^{(2)}, \boldsymbol{r}^{(1)}} = 0\} \cap \omega_{\boldsymbol{r}^{(1)}, \boldsymbol{r}^{(2)}}^{(1,2)}, \tag{25}$$

with $\alpha_{\boldsymbol{r}^{(1)}, \boldsymbol{r}^{(2)}} = (A_{\boldsymbol{r}^{(2)}}^{(2)} A_{\boldsymbol{r}^{(1)}}^{(1)})^T [A^{(3)}]_{1,1,\cdot}$ and $\beta_{\boldsymbol{r}^{(1)}, \boldsymbol{r}^{(2)}} = [A^{(3)}]_{1,1,\cdot}^T A_{\boldsymbol{r}^{(2)}}^{(2)} B_{\boldsymbol{r}^{(1)}}^{(1)} + [B^{(3)}]_{1,1}.$[4] The distribution of $\alpha_{\boldsymbol{r}^{(1)}, \boldsymbol{r}^{(2)}}$ characterizes the structure of the decision boundary and thus highlights the interplay between the layer parameters, layer topology, and the decision boundary. For example, in Fig. 2 the red line demonstrates how the weights characterize the curvature and cut positions of the decision boundary. We provide examples highlighting the impact on the angles of change in the architecture of the DN in Appendix A.5.

We provide a direct application of the above finding by providing a curvature characterization of the decision boundary. First, we propose the following result stating that the form of $\alpha$ and $\beta$ from (25) from a region to a neighbouring one alters only a single unit code at a some layer.

**Lemma 3.** *Upon reaching a region boundary, any edge as defined in Definition 3 must continue into a neighbouring region.*

This follows directly from continuity of the involved operator and enables us to study its curvature by comparing the edges of adjacent regions. In fact adjacent region edges connect at the region boundary by continuity, however their angle might differ, this angle defines the curviness of the decision boundary, which is defined as the collection of all the edges introduces by the last layer.

**Theorem 5.** *The decision boundary curvature/angle between two adjacent regions[5] $\boldsymbol{r}$ and $\boldsymbol{r}'$ is given by the following dihedral angle [KB38] between neighbouring $\alpha$ parameters as*

$$\cos(\theta(\boldsymbol{r}, \boldsymbol{r}')) = \frac{|\langle \alpha_{\boldsymbol{r}}, \alpha_{\boldsymbol{r}'} \rangle|}{\|\alpha_{\boldsymbol{r}}\| \|\alpha_{\boldsymbol{r}'}\|}. \tag{26}$$

### Acknowledgements

RB and RGB were supported by NSF grants CCF-1911094, IIS-1838177, and IIS-1730574; ONR grants N00014-18-12571 and N00014-17-1-2551; AFOSR grant FA9550-18-1-0478; DARPA grant G001534-7500; and a Vannevar Bush Faculty Fellowship, ONR grant N00014-18-1-2047. RC and BA were supported by NSF grant SCH-1838873 and NIH grant R01HL144683-CFDA.

## Footnotes

[1]The three subscripts of the slopes tensor $[A]_{k,r,d}$ correspond to output $k$, partition region $r$, and input signal index $d$. The two subscripts of the offsets/biases tensor $[B]_{k,r}$ correspond to output $k$ and partition region $r$.

[2] Orthogonal DN filters have the property that $\langle [A]_{k,r,\cdot}, [A]_{k',r',\cdot} \rangle = 0, \forall r, r', k \neq k'$.

[3] The saliency maps were linked to the filters in a matched filterbank in [BB18a, BB18b].

[4]The last layer is a linear transform with one unit, since we perform binary classification.

[5]For clarity, we omit the subscripts.

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
