[Supplementary Material]

Rank 1          Rank 2 Orthogonal          Rank 2

Figure 5: Examples of various weight constraints and their impact on the layer input space partitioning. (Left) A low-rank weight matrix leads to colinear cuts, with the cut spacing determined by the bias values. (Middle) An orthogonal weight matrix leads to orthogonal cuts. (Right) An arbitrary weight matrix leads to an unconstrained partition.

## A    Additional Geometric Insights

The *Laguerre distance* corresponds to the length of the line segment that starts at $x \in \mathcal{X}$ and ends at the tangent to the hypersphere with center $[A^{(\ell)}]_{k,r^\star,.}$ and radius $[\mathrm{rad}]_{k,r^\star}$ (see Fig. 1).

The *hyperplanar boundary* between two adjacent power diagram (PD) regions can be characterized in terms of the *chordale* of the corresponding hyperspheres [Joh60]. Doing so for all adjacent boundaries fully characterizes the region boundaries in simple terms of hyperplane intersections from [Aur87].

### A.1    Paraboloid $U$ Insights

A further characterization of the polytope boundary $\partial \mathbf{P}_k$ can be made by introducing the *paraboloid $U$* defined as $U(x) = \frac{1}{2}\|x\|_2^2$. Note that the slope of $\mathcal{E}_{k,r}$ is $\nabla \mathcal{E}_{k,r} = [A]_{k,r,.}$ and its offset is $-\frac{1}{2}\|[A]_{k,r,.}\|_2^2$. Notice that $\mathcal{E}_{k,r}$ defines an hyperplane in $\mathbb{R}^{D+1}$ *the hyperplane $\mathcal{E}_{k,r}$ is the tangent of the paraboloid $U$ at the point $[A]_{k,r,.}$.* We now highlight that the hyperplane and the paraboloid intersect at a unique point

$$U(x) \begin{cases} = \mathcal{E}_{k,r}(x) \iff x = [A]_{k,r,.} \\ > \mathcal{E}_{k,r}(x) \text{ , otherwice} \end{cases} \tag{27}$$

The faces of the $\mathbf{P}_k$ are the tangent of $U$ at the points given by $[A]_{k,r,.}, \forall r$ leading to

$$U(x) \begin{cases} = \mathbf{P}_k(x) \iff x \in \{[A]_{k,r,.}, \forall r\} \\ > \mathbf{P}_k(x) \text{ , otherwise} \end{cases} \tag{28}$$

The intersection of the hyperplane with the paraboloid produces an hypersphere (as seen in Fig. 1) that is characterized by the following propostion.

**Proposition 4.** *[Aur87] There is a bijective mapping between the hyperpshere in the input domain and the intersection of an hyperplane $\mathcal{E}$ in $\mathbb{R}^{D+1}$ with a paraboloid $U$ in $\mathbb{R}^{D+1}$.*

In fact, the projection of the intersection of the hyperplane and the paraboloid onto the input space forms a circle with radius corresponding to the shift of the hyperplane.

### A.2    Power Diagram Weight Constraints and Region Shapes

We highlight the close relationship between the layer slopes and offsets $A, B$ from (1), the layer polytope $\mathbf{P}$ from Proposition 2, and the boundaries of the layer PD from Theorem 2. In particular, we consider how one can alter or constrain the shape of the cells by constraining the weights of the layer.

**Example 1:** Constraining the layer weights to be such that $[A]_{k,r,d} = 1_{\{d=[i]_{k,r}\}}[\mathrm{cst}]_{k,r}$ for some integer $[i]_{k,r} \in \{1, \ldots, D\}$, $D = \dim(\mathcal{X})$, and arbitrary constant $[\mathrm{cst}]_{k,r}$ leads to an input PD with cell boundaries parallel to the input space basis vectors see Fig. 5. For instance, if the input space $\mathcal{X}$ is the Euclidean space $\mathbb{R}^D$ equipped with the canonical basis, then the Proposition 4 translates into having PD boundaries parallel to the axes.

**Example 2:** Constraining the layer weights to be such that $[A]_{k,r,d} = \pm[\mathrm{cst}]_{k,r}$ for some arbitrary constant $[\mathrm{cst}]_{k,r}$ leads to a layer-input power diagram with diagonal cell boundaries.[6]

**Theorem 6.** *Updating a single unit's parameters (slope or offset of the affine transform and/or the nonlinearity behavior) affects multiple regions' centroids and radii.*

The above result highlights the weight sharing concepts and implicit bias/regularization. In fact, most regions are tied together in terms of learnable parameters. Modifying a single region while leaving everything else the sameuntouched is not possible in general.

Table 1: We experiment the ability of the PD centroids to be close to their associated input. Recall that in a PD, the radii allows the centroids to move away from their region, which is not the case of a VD. We thus compare this centroid/region linkage with the following experiment. We extract the centroid induced by the region of a given input. In a VD setting, given that the regions are small, the centroids would be very close to the input and thus when computing the distance form this centroid to all inputs, the input used for the centroid generation would be the one closest to the centroid. This is not longer true in the PD setting where another input in another region might be closer to the centroids (due to the radii). We thus compare here the argmax position of the used input, a small number implies that the PD is close to a VD and that the centroids lie in or nearby their region.

| Layer | 1 | 2 | 3 | 4 | 5 | 6 | 7 | 8 | 9 |
|---|---|---|---|---|---|---|---|---|---|
| SmallConv (init.) | 0 | 215 | 10 | 1579 | 3458 | - | - | - | - |
| SmallConv (trained) | 0 | 31 | 2848 | 3511 | 1591 | - | - | - | - |
| LargeConv (init.) | 1 | 27 | 0 | 178 | 240 | 46 | 1534 | 2115 | 4644 |
| LargeConv (trained) | 0 | 0 | 0 | 472 | 3005 | 3763 | 4923 | 4280 | 4267 |
| Resnet (init.) | 0 | 1 | 19 | 17 | 17 | 9 | 15 | 20 | - |
| Resnet (trained) | 0 | 0 | 10 | 25 | 40 | 200 | 254 | 591 | - |

Figure 6: A) distribution of the log of the distances through the epochs (during leaning), B) through the layer (as the partitioning gets subdivided). The statistics are computed on the train set (blue) and test set (red). It is clear that deeper layers aim at refining only some regions, likely the ones hard to classify. This is shown by the tails becoming heavier. For additional figures see Fig. 10.

## A.3 Computational Complexity Properties

We highlight a key computational property of DNs contributing to their success. While the actual number of cells from a layer PD varies greatly depending on the parameters, the cell inference task always search over the maximum $R_{\mathrm{upper}}^{(\ell)} = R^{D(\ell)}$ number of cells as

$$\boldsymbol{r}^{(\ell)}(\boldsymbol{x}) = \underset{\boldsymbol{r} \in \{1,\dots,R\}^{D(\ell)}}{\arg\min} \|\boldsymbol{z}^{(\ell-1)}(\boldsymbol{x}) - \mu_{\boldsymbol{r}}^{(\ell)}\| - \mathrm{rad}_{\boldsymbol{r}}^{(\ell)}. \tag{29}$$

The computational and memory complexity of this task is $\mathcal{O}(R_{\mathrm{upper}}^{(\ell)}D(\ell-1))$. While approximations exist [ML09, AMN$^+$98, GSM03], we demonstrate how a MASO-induced PD is constructed in a way that it is parameter-, memory-, and computation-efficient.

**Lemma 4.** *A DN layer solves (29) with computational and memory complexity $\mathcal{O}(\log_{R^{(\ell)}}(R_{\mathrm{upper}}^{(\ell)})R^{(\ell)}D(\ell - 1)) = \mathcal{O}(D(\ell)R^{(\ell)}D(\ell-1))$ as opposed to $\mathcal{O}(R_{\mathrm{upper}}^{(\ell)}D(\ell-1)) = \mathcal{O}((R^{(\ell)})^{D(\ell)}R^{(\ell)}D(\ell-1))$ for an arbitrary PD.*

The entire DN then solves iteratively (29) for each layer.

**Theorem 7.** *An entire DN infers the cell associated to an input $\boldsymbol{x} \in \mathcal{X}$ with computational and memory complexity $\mathcal{O}(\sum_{\ell=1}^{L} D(\ell)R^{(\ell)}D(\ell - 1))$ as opposed to $\mathcal{O}(\sum_{\ell=1}^{L}(R^{(\ell)})^{D(\ell)}R^{(\ell)}D(\ell - 1))$ for an arbitrary hierarchy of Power Diagrams.*

The above practical result is crucial to the ability of DN layers to perform extremely fine grained input space partitioning without sacrificing computation time.

## A.4 Further Results on Empirical Region Characterization

See Figure 6 and Table 1.

## A.5    Further Results on Decision Boundary Curvature

Recall that a neighbouring region implies a change of a single code unit in $r^{(1)}$. Denote without loss of generality the changed code index by $d' \in \{1, \ldots, D^{(1)}\}$. The other $D^{(1)} - 1$ codes of $r^{(1)}$ remain the same. When dealing with $R = 2$ nonlinearities, this implies that $[r]_{d'}$ changes from a 1 to a 2 for those two neighbouring regions or vice-versa. Let denote by $r$ the case with a 1 at this index $d'$ and by $r'$ the case with a 2 at this index $d'$. With those notations, we can derive some special cases of the distance formula (26) for some DN topologies.

**Proposition 5.** *In a 2-layer DN with ReLU and orthogonal first layer weights $W^{(1)}$, we have*

$$\cos(\theta(r, r')) = \left( \frac{|[W^{(2)}]_{1,d'}| \|[W^1]_{d',.}\|_2}{\sum_{d \neq d'}^{D^{(1)}} 1_{\{[r']_d = 2\}} |[W^{(2)}]_{1,d}| \|[W^{(1)}]_{d,.}\|} + 1 \right)^{-1} \in (0, 1) \tag{30}$$

In the above result, the angle between the adjacent region segments of the two regions with code $r$ and $r'$ is characterized w.r.t. the first layer weights when consider a two layer DN. From the above formula it is clear that reducing the norm of the weights alone does not impact the angles. However, we have the following result when in additional of orthogonality we assume similar norms for the layer weights.

**Proposition 6.** *When the first layer weights have similar norms $\|[W^{(1)}]_{k,.}\| \approx \|[W^{(1)}]_{k',.}\|_2, k \neq k'$, regions of the input space in which the amount of firing ReLU obtained by $\|r^{(1)}(x)\|_0$ is small will have greater decision boundary curvature than regions with most ReLU firing simultaneously.*

As a result, a ReLU based DN will have different behavior at different regions of the space due to the 0, 1 ReLU weights regardless of the constraints one can put on the layer weights.

# B  Additional Visualizations

Figure 7: Examples of the partitioning and subdivision happening layer after layer. Each unit also introduces a path in the input space which is depicted below the current partitioning with the highlighted path linked via a dotted line.

Figure 8: Visual depiction of PD division (Sec. 4.1) for a 2-layer DN with 3 units at the first layer (leading to 4 regions) and 8 units at the second layer with random weights and biases. The colors are the DN input space partitioning with respect to the first layer. Then for each color (or region) the layer1-layer2 defines a specific PD that will sub-divide this aforementioned region (this is the first row) where the region is colored and the PD is depicted for the whole input space. Then this sub-division is applied onto the first layer region only as it only sub-divides its region (this is the second row on the right). And finally grouping together this process for each of the 4 region, we obtain the layer-layer 2 space partitioning (second row on the left).

Figure 9: Examples of the partitioning and subdivision happening layer after layer. Each unit also introduces a path in the input space which is depicted below the current partitioning with the highlighted path linked via a dotted line.

Figure 11: Extension of Fig. 3: Examples of the centroids of the PD regions that contain an input $x$ for the smallCNN architecture

Figure 10: Additional depiction of the distances distribution.

In a standard Voronoi partition, the centroids alone describe the regions and are guaranteed to be inside their region. In a PD, the interplay between radii and centroids can make the centroids depart from their respective regions, removing the visual interpretation that the Voronoi partitions provide. Nevertheless, Figures 11,3 provide hints at how through depth, the partitioning moves from a close to Voronoi diagram one (with centroids close to the input) to a radius based partitioning. This could open the door to either new constrained DNs with more interpretable centroids (as described at the end of Sec. 3.2), or novel geometric visualization of PDs.

## C   DN Architectures

For all experiments, we used the standard test set for each dataset and a train/valid split of $15\%$. The batch size was $32$ for the distance experiment and $64$ for the other, and we trained for $150$ epochs. In all cases, the Adam optimizer was used.

### C.1   LargeConv

```
Conv2D(Number Filters=96, size=3x3, Leakiness=0.01))

Conv2D(Number Filters=96, size=3x3, Leakiness=0.01))

Conv2D(Number Filters=96, size=3x3, Leakiness=0.01))

Pool2D(2x2)
```

```
Conv2D(Number Filters=192, size=3x3, Leakiness=0.01))

Conv2D(Number Filters=192, size=3x3, Leakiness=0.01))

Conv2D(Number Filters=192, size=3x3, Leakiness=0.01))

Pool2D(2x2)

Conv2D(Number Filters=192, size=3x3, Leakiness=0.01))

Conv2D(Number Filters=192, size=1x1, Leakiness=0.01))

Conv2D(Number Filters=Number Classes, size=1x1, Leakiness=0.01))

GlobalPool2D(pool_type='AVG'))
```

## C.2    SmallConv

```
Conv2D(Number Filters=32, size=5x5, Leakiness=0.01))

Pool2D(2x2)

Conv2D(Number Filters=64, size=3x3, Leakiness=0.01)

Conv2D(Number Filters=64, size=3x3, Leakiness=0.01)

Pool2D(2x2)

Conv2D(Number Filters=128, size=3x3, Leakiness=0.01)

Conv2D(Number Filters=Number Classes, size=1x1)

GlobalPool2d(pool_type='AVG')
```

# D    Distance To Boundary

The distance from a point to the partition boundary as given in (20) is obtained as follows. Recall that the DN input space partitioning is successively refined at each layer. That is, when adding laeyr $\ell$, the input space partition is refined but do not alter the previously built partition $\Omega^{(1,\dots,\ell-1)}$. Also, recall that each of the partition boundary correspond to the zero set of each of the layer unit (recall (22)) and this for each layer. When searching over all layers and all units of each layer for the smallest distance between the point and the affine mapping zero set, we thus obtain the smallest distance from a point to the DN input space partitioning.

# E    Proofs

## E.1    Proof of Lemma 1: Single Unit Projection

This follows from [Aur87] Lemma 4 (page 84) that demonstrates that the boundaries in the input space $\mathcal{X}$ defining the regions of the unit PD are the vertical projections of the polytope $(\mathcal{P}_k)$ face intersections defined as $\mathcal{E}_{k,r}(\mathcal{X}) \cap \mathcal{E}_{k,r*}(\mathcal{X})$ for neighbouring faces $r$ and $r^*$.

## E.2    Proof of Lemma 2: Single Layer Projection

This follows from Sec. 3.2 in which it is demonstrated that the boundaries of a single unit PD is obtained by vertical projection of the polytope edges. In the layer case, the edges of $\mathbf{P}$ correspond to all the points in the input space $\mathcal{X}$ s.t. $\boldsymbol{z}$ belongs to an edge of at least one of the polytopes $\mathcal{P}_k, \forall k$ making up $\mathbf{P}$. The layer PD having for boundaries the union of all the per unit PD boundaries, it follows directly that the vertical projection of the edges of $\mathbf{P}$ form the layer PD boundaries.

## E.3    Proof of Theorem 1: Single Layer Power Diagram

Consider the case of 2 units.

**Lemma 5.** *The layer input space of the $[1^{st}, 2^{nd}]$-MASO units at layer $l$ is a weighted Voronoi Diagram with a maximum of $R^\ell \times R^\ell$ regions, centroids $\boldsymbol{A}\{[t]_1, [t]_2\} = [\boldsymbol{A}]_{1,[t]_1,\cdot} + [\boldsymbol{A}]_{2,[t]_2,\cdot}$, and biases $b\{[t]_1, [t]_2\} = [b]_{1,[t]_1} + [b]_{2,[t]_2} - 2\langle[\boldsymbol{A}]_{1,[t]_1,\cdot}, [\boldsymbol{A}]_{2,[t]_2,\cdot}\rangle$.*

*Proof.* Denote by $\mathcal{V}([t]_1)$ a region of unit 1 and $\mathcal{V}([t]_2)$ a region of unit 2 and $\mathcal{V}([t]_1, [t]_2)$ the joint region of both units. Then we have

$$\mathcal{V}([t]_1, [t]_2) = \mathcal{V}([t]_1) \cap \mathcal{V}([t]_2)$$

$$= \{\boldsymbol{x} \in \mathbb{R}^D | \arg\max_i \langle \boldsymbol{x}, [\boldsymbol{A}]_{1,i,\cdot} \rangle + [B]_{1,i} = [t]_1\} \cap \{\boldsymbol{x} \in \mathbb{R}^D | \arg\max_j \langle \boldsymbol{x}, [\boldsymbol{A}]_{2,j,\cdot} \rangle + [B]_{2,j} = [t]_2\}$$

$$= \{\boldsymbol{x} \in \mathbb{R}^D | \arg\min_i \|\boldsymbol{x} - [\boldsymbol{A}]_{1,i,\cdot}\|^2 + [b]_{1,i} = [t]_1\}$$

$$\cap \{\boldsymbol{x} \in \mathbb{R}^D | \arg\min_j \|\boldsymbol{x} - [\boldsymbol{A}]_{2,j,\cdot}\|^2 + [b]_{2,j} = [t]_2\}$$

$$= \{\boldsymbol{x} \in \mathbb{R}^D | \arg\min_{i,j} \|\boldsymbol{x} - [\boldsymbol{A}]_{1,i,\cdot}\|^2 + [b]_{1,i} + \|\boldsymbol{x} - [\boldsymbol{A}]_{2,j,\cdot}\|^2 + [b]_{2,j} = ([t]_1, [t]_2)\}$$

$$= \{\boldsymbol{x} \in \mathbb{R}^D | \arg\min_{i,j} 2\|\boldsymbol{x}\|^2 + \|[\boldsymbol{A}]_{1,i,\cdot}\|^2 - 2\langle \boldsymbol{x}, [\boldsymbol{A}]_{1,i,\cdot} \rangle$$

$$+ [b]_{1,i} + \|[\boldsymbol{A}]_{2,j,\cdot}\|^2 - 2\langle \boldsymbol{x}, [\boldsymbol{A}]_{2,j,\cdot} \rangle + [b]_{2,j} = ([t]_1, [t]_2)\}$$

$$= \{\boldsymbol{x} \in \mathbb{R}^D | \arg\min_{i,j} 2\|\boldsymbol{x}\|^2 - 2\langle \boldsymbol{x}, [\boldsymbol{A}]_{1,i,\cdot} + [\boldsymbol{A}]_{2,j,\cdot} \rangle + \|[\boldsymbol{A}]_{1,i,\cdot}\|^2$$

$$+ \|[\boldsymbol{A}]_{2,j,\cdot}\|^2 + [b]_{1,i} + [b]_{2,j} = ([t]_1, [t]_2)\}$$

$$= \{\boldsymbol{x} \in \mathbb{R}^D | \arg\min_{i,j} \|\boldsymbol{x} - ([\boldsymbol{A}]_{1,i,\cdot} + [\boldsymbol{A}]_{2,j,\cdot})\|^2 + [b]_{1,i} + [b]_{2,j} - 2\langle [\boldsymbol{A}]_{1,i,\cdot}, [\boldsymbol{A}]_{2,j,\cdot} \rangle = ([t]_1, [t]_2)\}$$

$$= \{\boldsymbol{x} \in \mathbb{R}^D | \arg\min_{i,j} \|\boldsymbol{x} - \boldsymbol{A}\{i,j\}\|^2 + b\{i,j\} = ([t]_1, [t]_2)\},$$

where $b\{i,j\} = \|[\boldsymbol{A}]_{1,i,\cdot}\|^2 + 2[B]_{1,i} + \|[\boldsymbol{A}]_{2,j,\cdot}\|^2 + 2[B]_{2,j} + 2\langle [\boldsymbol{A}]_{1,i,\cdot}, [\boldsymbol{A}]_{2,j,\cdot} \rangle$ and $\boldsymbol{A}\{i,j\} = [\boldsymbol{A}]_{1,i,\cdot} + [\boldsymbol{A}]_{2,j,\cdot}$.

In the above, the second step was obtained by noting that adding a constant to both sides do not change the region allocation of the inputs. As such, we add the $\ell_2$ norm of the input to express the inner product plus bias as a norm plus bias. The additional correlation coefficients introduced by the norm are then subtracted by modifying of the bias leading to $B$ becoming $b$. As a result the second step expresses the inner product plus bias into a norm plus altered bias. Following this, the next steps simply involve calculus to obtain a single expression for both regions simultaneously.

$\square$

For the $D$ units case, we recursively apply Lemma 5.

### E.4 Proof of Theorem 2: Single Layer Power Diagram

We first derive a preliminary result in which a layer follows an affine transformation to then generalize by considering how each region of the previously built partitioning transforms the inputs lying in it linearly.

**Input Space Partitioning of a Single Layer Following an Affine Transform**   Consider a layer with input an affine transformation of $\boldsymbol{x} \in \mathcal{X}$ as $G\boldsymbol{x} + \boldsymbol{h}$ where $G$ is an arbitrary matrix and $\boldsymbol{h}$ an arbitrary vector. We consider this affine transformation as a linear DN layer. We now express the layer PD with respect to the input space as $\Omega^{(2)}(\mathcal{X}^{(0)})$. Define the centroids and radius as

$$\mu_{\boldsymbol{r}^{(2)}}^{(1 \leftarrow 2)} = G^\top \sum_{k=1}^{D^{(2)}} [A^{(2)}]_{k,[\boldsymbol{r}^{(2)}]_k,\cdot} = G^\top \mu_{\boldsymbol{r}^{(2)}}^{(2)} \tag{31}$$

$$\mathrm{rad}_{\boldsymbol{r}^{(2)}}^{(1 \leftarrow 2)} = -\|G^\top \mu_{\boldsymbol{r}^{(2)}}^{(1 \leftarrow 2)}\|^2 - 2\mu_{\boldsymbol{r}^{(\ell)}}^{(1 \leftarrow 2)\top} \boldsymbol{h} - 2\sum_{k=1}^{D^{(2)}} [B^{(2)}]_{k,[\boldsymbol{r}]_k} \tag{32}$$

where $\mu_{\boldsymbol{r}^{(2)}}^{(2)}$ is as defined in Theorem 2.

**Lemma 6.** *The input space partitioning of a 2-layer DN with first layer linear is given by*
$\Omega^{(1,2)}(\mathcal{X}^{(0)}) = PD(\mathcal{X}; \{(\mu_{\boldsymbol{r}}, \mathrm{rad}_{\boldsymbol{r}}), \forall \boldsymbol{r}\}).$

**Lemma 7.** *The layer input space of the $[1^{st}, 2^{nd}]$-MASO units at layer $l$ is a weighted Voronoi Diagram with a maximum of $R^\ell \times R^\ell$ regions, centroids $\boldsymbol{A}\{[t]_1, [t]_2\} = G^T[\boldsymbol{A}]_{1,[t]_1,\cdot} + G^T[\boldsymbol{A}]_{2,[t]_2,\cdot}$, and biases $b\{[t]_1, [t]_2\} = [b]_{1,[t]} + [b]_{2,[t]_2} - 2\langle G^T[\boldsymbol{A}]_{1,[t]_1,\cdot}, G^T[\boldsymbol{A}]_{2,[t]_2,\cdot} \rangle.$*

*Proof.*

$$\mathcal{V}([t]_1, [t]_2) = \mathcal{V}([t]_1) \cap \mathcal{V}([t]_2)$$

$$= \{\boldsymbol{x} \in \mathbb{R}^D | \arg\max_i \langle G\boldsymbol{x} + h, [\boldsymbol{A}]_{1,i,\cdot}\rangle + [B]_{1,i} = [t]_1\}$$

$$\cap \{\boldsymbol{x} \in \mathbb{R}^D | \arg\max_j \langle G\boldsymbol{x} + h, [\boldsymbol{A}]_{2,j,\cdot}\rangle + [B]_{2,j} = [t]_2\}$$

$$= \{\boldsymbol{x} \in \mathbb{R}^D | \arg\max_i \langle \boldsymbol{x}, G^T[\boldsymbol{A}]_{1,i,\cdot}\rangle + \langle [\boldsymbol{A}]_{1,i,\cdot}, h\rangle + [B]_{1,i} = [t]_1\}$$

$$\cap \{\boldsymbol{x} \in \mathbb{R}^D | \arg\max_j \langle \boldsymbol{x}, G^T[\boldsymbol{A}]_{2,j,\cdot}\rangle + \langle [\boldsymbol{A}]_{2,j,\cdot}, h\rangle + [B]_{2,j} = [t]_2\}$$

$$= \{\boldsymbol{x} \in \mathbb{R}^D | \arg\min_i \left\| \boldsymbol{x} - G^T[\boldsymbol{A}]_{1,i,\cdot} \right\|^2 + [b]_{1,i} = [t]_1\}$$

$$\cap \{\boldsymbol{x} \in \mathbb{R}^D | \arg\min_j \left\| \boldsymbol{x} - G^T[\boldsymbol{A}]_{2,j,\cdot} \right\|^2 + [b]_{2,j} = [t]_2\}$$

$$= \{\boldsymbol{x} \in \mathbb{R}^D | \arg\min_{i,j} \left\| \boldsymbol{x} - G^T[\boldsymbol{A}]_{1,i,\cdot} \right\|^2 + [b]_{1,i} + \left\| \boldsymbol{x} - G^T[\boldsymbol{A}]_{2,j,\cdot} \right\|^2 + [b]_{2,j} = ([t]_1, [t]_2)\}$$

$$= \{\boldsymbol{x} \in \mathbb{R}^D | \arg\min_{i,j} 2\left\| \boldsymbol{x} \right\|^2 + \left\| G^T[\boldsymbol{A}]_{1,i,\cdot} \right\|^2 - 2\langle \boldsymbol{x}, G^T[\boldsymbol{A}]_{1,i,\cdot}\rangle + [b]_{1,i}$$

$$+ \left\| G^T[\boldsymbol{A}]_{2,j,\cdot} \right\|^2 - 2\langle \boldsymbol{x}, G^T[\boldsymbol{A}]_{2,j,\cdot}\rangle + [b]_{2,j} = ([t]_1, [t]_2)\}$$

$$= \{\boldsymbol{x} \in \mathbb{R}^D | \arg\min_{i,j} 2\left\| \boldsymbol{x} \right\|^2 - 2\langle \boldsymbol{x}, G^T[\boldsymbol{A}]_{1,i,\cdot} + G^T[\boldsymbol{A}]_{2,j,\cdot}\rangle$$

$$+ \left\| G^T[\boldsymbol{A}]_{1,i,\cdot} \right\|^2 + \left\| G^T[\boldsymbol{A}]_{2,j,\cdot} \right\|^2 + [b]_{1,i} + [b]_{2,j} = ([t]_1, [t]_2)\}$$

$$= \{\boldsymbol{x} \in \mathbb{R}^D | \arg\min_{i,j} \left\| \boldsymbol{x} - (G^T[\boldsymbol{A}]_{1,i,\cdot} + G^T[\boldsymbol{A}]_{2,j,\cdot}) \right\|^2 + [b]_{1,i} + [b]_{2,j}$$

$$- 2\langle G^T[\boldsymbol{A}]_{1,i,\cdot}, G^T[\boldsymbol{A}]_{2,j,\cdot}\rangle = ([t]_1, [t]_2)\}$$

$$= \{\boldsymbol{x} \in \mathbb{R}^D | \arg\min_{i,j} \left\| \boldsymbol{x} - G^T \boldsymbol{A}\{i,j\} \right\|^2 + b\{i,j\} = ([t]_1, [t]_2)\},$$

where $b\{i,j\} = [b]_{1,i} + [b]_{2,j} - 2\langle G^T[\boldsymbol{A}]_{1,i,\cdot}, G^T[\boldsymbol{A}]_{2,j,\cdot}\rangle$ and $\boldsymbol{A}\{i,j\} = [\boldsymbol{A}]_{1,i,\cdot} + [\boldsymbol{A}]_{2,j,\cdot}$. and $[b]_{1,i} = -\|G^T[\boldsymbol{A}]_{1,i,\cdot}\|^2 - 2\langle [\boldsymbol{A}]_{1,i,\cdot}, h\rangle - 2[B]_{1,i}$. We thus have $b\{i,j\} = -2B\{i,j\} - 2A\{i,j\}^T h - \|G^T A\{i,j\}\|^2$. □

### E.5   Proof of Theorem 4

The polynomial is defined as

$$\prod_{\ell=1}^{L} \prod_{k=1}^{D(\ell)} (z_k^{(\ell)}(\boldsymbol{x}))$$

where we abbreviate $(z_k^{(\ell)} \circ \cdots \circ \boldsymbol{z}^{(1)})(\boldsymbol{x})$ by $z_k^{(\ell)}(\boldsymbol{x})$. As such, for each $\ell$, it is the product of $D(\ell)$ continuous piecewise linear functions thus making their product continuous piecewise polynomial function of order $D(\ell)$. Each of those are then multiplied together for each $\ell$ leading to the final order of the partitioning polynomial being of $\prod_{\ell=1}^{L} D(\ell)$. When using nonlinearities such as leaky-ReLU, the unit input space boundary is reached exactly when the unit output is 0. Following the polynomial construction, the product of the units will be 0 at the points in the input space for which at least one of the units will be 0 meaning that it lies on the region boundary of one of the region of the DN input space partitioning.

## Footnotes

[6]Note that, while in Example 1 each per unit $k$, per cell $r$ weight was constrained to contain a single nonzero element such as $(0, 0, c, 0)$ for $D = 4$, Example 2 fills the weight vector with a single constant but varying signs such as $(+c, -c, +c, -c)$.