[Reviews · NeurIPS 2019]

Reviewer 1



This paper studies the geometry of deep networks with piecewise affine and convex nonlinearities using the max affine spline operator (MASO) framework (BB18a, BB18b). The authors establish a connection between input space partitioning of a neural network and power diagrams. They present analytical formulas for the power diagrams and decision regions. The paper is well-written, and presents interesting theoretical results. It would be great if the authors can provide more details on the computational aspects. Please see below for detailed comments/questions. (0) It is obvious that deep neural networks with relu activations (and more generally MASOs) are piecewise linear functions where the input partitionings are convex polytopes (according to which max is active). Can you please clarify how does the power diagram notation advance this knowledge ? (1) In eq (1), it looks like the '.' (dot) in the notation [A^(l)]_{k,r,.} is not defined. Could you please clarify this (2) On page (2), is the notation z^{l}(x) used to explicitly state its dependence to the input x ? Is there any difference between x and bold case x ? (3) Can you please clarify what you mean by 'orthogonal filters' on line 158, page 5 ? (4) It would be great if the authors can provide details on computing the centroids and radii for all partitionings. What is the largest size that can be tackled ? Can they be approximated, e.g. by Monte Carlo ? (5) Can you explain how the centroids can be used in semi-supervised learning ? (line 216 page 7) (6) The proof of Theorem 4 doesn't appear to be included in the supplementary file (7) Can you please summarize how this work differs from the earlier works [BB18a, BB18b] ? (8) Notational issues in the supplementary material: In the proof of Theorem 1, what does the calligraphic v in V([t}_1) stand for ? Is the lower-case b same as upper-case B ? (9) Proof of Lemma 6 needs clarification in most of the steps. In particular, how does the second inequality follow ? (10) Can you provide the precise statement of the result from [Joh60] and its details (page number, theorem number etc.) ?

Reviewer 2



The shows that layered feed forward neural networks that are comprised for affine transforms and convex nonlinearities (e.g. RELU) can be seen as implementing a series of divisions of the input space whereby the division takes the form of a power diagram (a generalization of Voronoi tessellations) . This is proven by analysis of single units, analysis of single layers and analysis of layer-to-layer transitions. The work hinges on the description of layers as implementing a max operations over affine transforms (max affine spline operators, MASO) which have apparently been studied in this context previously. The use of power diagrams to describe MASO output in this context seems new. The paper goes on show some analysis results on the log-distances of training points in the CIFAR dataset from their power diagram centroids at different network layers. Finally, there is some analysis of decision boundary curvature. Figure 3 seems to show that centroid ad deeper and deeper layers which match a specific input points become less descriptive of that point, apparently due to larger and larger radii (regions are described by centroids and radii). This results seems somewhat counter intuitive and seems to hint that this the centroids do not behave in a simple comprehensible manner. Perhaps the figure should be better explained in the paper. There is no analysis of the neural network training processes or how the power diagram representation might help in understanding a network training session. The man theoretical results provides a comprehensive description of what the most popular basing network architecture can do. It would be interesting to see if that result could somehow be used to achieve better performance or training efficiency of such basic networks. There is no summary / concluding remarks and no code is provided (although there is evidence that code was used)

Reviewer 3



This paper uses a computational geometry approach to analyze the region decomposition of neural networks, which leads to power diagrams. Propositions and theorems in the paper however are not fundamentally new, following straightforwardly from the geometry approach. The authors nicely observe that their centroids can be efficiently computed via backpropagation and apply the analysis to power diagrams to understand neural networks. Visualizing the region centroids and averaged distance to boundaries reals interesting information, which are however sort of expected, e.g. distance decreases with more layers to subdivide input regions. Results and their interpretation in the paper are still very preliminary. Despite this, the paper advances in the promising direction of geometrically interpreting neural networks; it is to my surprise that contents in the paper has not been thoroughly studied in the past. I think conclusions in the paper will serve as the starting point that many future work in the direction will benefit from. Ideas in the paper such as considering curvature at decision boundary looks promising and may worth further study. Exposition: While the idea of the paper is clear from the high level, exact details of notations and equations in the paper are hard to follow in its current exposition. Notations in the paper sometimes is too complicated. E.g. f^{(1->l)}_{k} in Eq. 18 is not explicitly explained and I have to guess the meaning of superscripts. epsilon_{k,2}^(l-1) in Eq. 21 is not explained, and should the superscript be (l) or (l-1)? It should also be explained where Eq. 21 comes from. Eq. 20: why left hand side is squared distance? This equation is not explained. Typo: Line 333: hve -> have

[Author Response · NeurIPS 2019]

**Response to the Reviewers of "The Geometry of Deep Networks: Power Diagram Subdivision"**

We thank the reviewers for their careful reading, concrete suggestions, and interest in our manuscript. Responses to your Detailed Comments are given below. If our paper is accepted, the Python/TensorFlow code for all the experiments and the figures will be provided in a Github repository; this will clarify the computational details.

**Reviewer 1. [0]** Our power diagram (PD) framework exposes and advances the knowledge of the geometry of deep networks (DNs) in several ways. First, we demonstrate that MASO DN's convex regions are formed by an elegant subdivision process that we describe analytically in closed form. Second, our formulation exposes the roles played by the various parameters of the layers (the weights and biases) and the PDs' centroids and radii and opens the door to new computational geometric approaches to understanding and interpreting DNs. **[1]** We will clarify below (1) that $[A^{(\ell)}]_{k,r,\cdot}$ represents the vector containing all the values of the last dimension. **[2]** We will clarify in the revised Introduction that $x$ and $z^{(\ell)}$ represent either vectors or tensors depending on the context/layer and that boldface $\boldsymbol{z}^{(\ell)}$ and $\boldsymbol{x}$ represent the flattened versions of $z^{(\ell)}$ and $x$, respectively. **[3]** We will clarify in the text that "orthogonal" means $\langle [A]_{k,r,\cdot}, [A]_{k',r',\cdot} \rangle = 0, \forall r, r', k \neq k'$. **[4]** We will add below (18) the following "Remark: The centroid computation corresponds directly to the backpropagation algorithm (18) and thus can be computed precisely (up to roundoff error) and efficiently with same computational cost as a forward pass through the DN." **[5]** We will add just after the reference on line 216: "in which $\|\mu_{\boldsymbol{x}}^{(1 \leftarrow L)} - \boldsymbol{x}\|$ is used as the unsupervised loss." **[6]** We will add the proof of Theorem 4 to the Supplementary Materials and augment it with additional insights on the polynomial and its use to characterize the input space partitioning. **[7]** Previous work [BB18a, BB18b] has not characterized a DN's partitioning nor studied its construction through depth. Instead, [BB18a, BB18b] focused on the affine mappings that are applied on each of the partition regions. We will clarify this in paragraph 3 of the Introduction. **[8]** We will add the definition for $\mathcal{V}([t]_1)$ in the proof of Theorem 1; it is simply shorthand to denote the region generated by unit $[t]_1$; similarly, $b$ is shorthand that contains the elements of $A$ and $B$ as $b\{i, j\} = \|[A]_{1,i,\cdot}\|^2 + 2[B]_{1,i} + \|[A]_{2,j,\cdot}\|^2 + 2[B]_{2,j} + 2\langle [A]_{1,i,\cdot}, [A]_{2,j,\cdot} \rangle$. **[9]** We will add in the proof to Lemma 6 an explanation of each step. The second equality is derived by rewriting the inner product plus bias as a norm minus the remaining elements such that the equality holds. **[10]** We will add information regarding the theorem from [Joh60].

**Reviewer 2. [0]** We will add the following discussion around Fig. 3 and in the Supplementary Material regarding interpretability: In a Voronoi diagram, the centroids alone fully describe the partition regions and are guaranteed to lie inside. For a power diagram (PD), the interplay between the radii and centroids can make the centroids move out of their respective regions, complicating the visual interpretation. Interestingly, Fig. 3 suggests how, through depth, the partitioning moves from a near Voronoi partitioning, with centroids close to their associated regions (as measured by the small distances from the centroids to the input data points lying in each region) to a PD that relies heavily on the radii (as measured by the large distances from the centroids to the input data points lying in each region). This new understanding could open the door to novel DN constraints that provide more interpretable centroids (as described at the end of Sec. 3.2). **[1]** In the revised experiments section and Supplementary Materials, we will detail the training procedures (code will also be made publicly available). We will also make more explicit how training evolves the centroids and distances to the region boundaries (Figs. 3 and 4). **[2]** While the primary goal of the paper is to fully characterize the DN input space partitioning, we have provided some direction regarding how to use the results for (i) constraining the weights of the DN to impose specific PD region geometries (Appendix A.2); (ii) semi-supervised learning (end of Section 4.2); (iii) constraining the PD to be a VD for enhanced interpretability (end of Section 3.2); (iv) analysis of how the decision boundary curvature is constrained based on the DN topology and how it can be computed via a differentiable measure, enabling novel regularization techniques (Section 5.2). Unfortunately, due to the page limitation, there is no space to pursue any of these directions in greater detail. We plan to publish results in these directions in future papers.

**Reviewer 3. [0]** We will strive to make our notation and the connections more clear in the revised paper. One reason for the complications in the current notation is that we have striven to connect the standard computational geometry notation with the standard DN notation. For example, we believe that the region shape characterization (Appendix A.2) and decision boundary curvature (Section 5.2 and Appendix A.5) are more interpretable with the current notation, for the deep learning community. However, as you pointed out, some notation was not introduced nor sufficiently explained; we will add those in the revised paper (such as for (18)). **[1]** We will add a sentence setting up the notation and setting before (21) and correct $\ell$ to $\ell - 1$. **[2]** Indeed, the left-hand side should not have been squared; we will correct this. We will also add a paragraph in the Supplementary Material (due to the lack of space in the main text) to fully explain the distance and add a reference along with a short descriptive sentence in the main text. **[3]** The typo on line 333 will be corrected.

[Meta-Review · NeurIPS 2019]

This paper provides a power diagram for the decision surface of relu networks (where power diagram is as with kmeans and voronoi cells). If the paper is not accepted due to the competitiveness of neurips this year, i urge the authors to develop further consequences of their construction (e.g., ways it aids in other questions related to deep networks).